# Peptide Stapling Applied to Antimicrobial Peptides

**DOI:** 10.3390/antibiotics12091400

**Published:** 2023-09-02

**Authors:** Ana Laura Pereira Lourenço, Thuanny Borba Rios, Állan Pires da Silva, Octávio Luiz Franco, Marcelo Henrique Soller Ramada

**Affiliations:** 1Programa de Pós-Graduação em Ciências Genômicas e Biotecnologia, Universidade Católica de Brasília, Brasília 71966-700, Brazil; 2S-Inova Biotech, Pós-Graduação em Biotecnologia, Universidade Católica Dom Bosco, Campo Grande 79117-900, Brazil; 3Programa de Pós-Graduação em Gerontologia, Universidade Católica de Brasília, Brasília 71966-700, Brazil

**Keywords:** antimicrobial peptides, stapling, all-hydrocarbon stapling, cyclization

## Abstract

Antimicrobial peptides (AMPs) are considered a promising therapeutic approach against multi-drug resistant microorganisms. Besides their advantages, there are limitations to be overcome so that these molecules can become market competitive. One of the biggest limitations is proteolytic susceptibility, which could be overcome by structural modifications such as cyclization, especially for helix-constraining strategies. Over the years, many helix stabilization techniques have arisen, such as lactam-bridging, triazole-based, N-alkylation and all-hydrocarbon stapling. All-hydrocarbon stapling takes advantage of modified amino acid residues and olefinic cross-linking to constrain peptide helices. Despite being a well-established strategy and presenting efficient stability results, there are different limitations especially related to toxicity. In this review, recent studies on stapled AMPs for antimicrobial usage are explored with the aim of understanding the future of these molecules as putative antimicrobial agents.

## 1. Introduction

Diseases and infections affecting humans, animals and plants are mostly managed by antimicrobial agents. Unfortunately, these antimicrobial agents, which are critical tools in our society, are becoming ineffective when facing resistant strains. The emergence of multi-drug resistant microorganisms has been a human and animal health threat since the discovery of resistant strains [1,2,3]. During the last three decades, antimicrobial peptides (AMPs) have demonstrated potential as antibiotic drugs able to face this global problem. Through wide-ranging research, natural and non-natural AMPs have been observed to kill bacteria through many mechanisms, including membranolytic and non-membranolytic modes of action, although the specific molecular targets for most peptides are still poorly described [4,5]. However, some physicochemical properties such as positive charge and amphiphilic properties have been widely associated with most peptides’ antimicrobial activity [3].

Despite being interesting candidates, linear AMPs are usually unstable molecules that can rapidly undergo proteolytic degradation. Apart from their instability, they cause nonspecific membrane toxicity, presenting cytotoxic activity at low concentrations. These are by far the main limitations holding back the development of AMP-based antibiotic drugs [6,7,8]. Therefore, chemical modifications of AMPs are essentially oriented to improve their stability and antimicrobial activity and decrease cytotoxicity.

Several chemical modifications can be made to peptide structures to improve their properties, whether chemical, physical or biological. Nowadays, the main approaches proposed are insertion of D-amino acids, PEGylation, acetylation, dimerization, lipidation and cyclization (Table 1) [7,9,10,11,12,13,14,15]. Peptide cyclization is one of the most promising techniques to improve their stability and antimicrobial activity. Considering that AMPs’ activity is mostly due to membrane disruption, restraining peptide conformation flexibility may facilitate insertion in the membrane by diminishing the entropic barrier related to the loss of freedom of molecular conformation [16,17]. Nevertheless, further in this review, flexibility aspects are discussed as being an interesting option for more efficient interactions.

Cyclization can be obtained via head-to-tail, side chain-to-head, side chain-to-tail or side chain-to-side chain bonding (Figure 1A) through different strategies [18,19]. Particularly for α-helical AMPs, but not exclusively [20], a cyclization approach called stapling has been used to constrain one face of the α-helix. Briefly, peptide stapling is characterized by cyclization between two side chain residues, resulting in a loss of peptide flexibility, which produces a more stable and less proteolysis-susceptible molecule (Figure 1B–E) [9,17,18].

Many stapling techniques attempt to constrain helical conformation in short peptides, tethering amino acid residues at *i*, *i* + 4 or *i, i* + 7 positions (where *i* is a given amino acid position). These stapling positions are more usual because naturally occurring proteins generally have their α-helices stabilized by intramolecular hydrogen bonds between the carbonyl and the amide groups of *i* and *i* + 4 residues, making the helix present around 3.6 residues per turn. Therefore, such positions are able to extend the clamp to one (*i*, *i* + 4) (Figure 1B) or two turns (*i, i* + 7) of the α-helix (Figure 1C) in linear peptides [9,18,19,21,22]. Nonetheless, stapled helices at *i, i + 11*, for three turns and *i, i + 3* for a single turn are also viable in special cases [23,24].

Over the years, many helix stabilization strategies have arisen, from lactam bridging between the natural side chains of appropriately spaced amino acids to approaches using non-natural amino acids, including α, α-disubstituted amino acids, and many others like thiol-based, azide-alkyne cycloaddition, triazole and hydrocarbon stapling [18,25,26]. However, describing stapling techniques is not the focus of this review; such methodologies can be found reviewed elsewhere [27,28,29,30]. Here, we present the ways in which stapled peptides have been explored for antimicrobial usage in recent years.

## 2. Antimicrobial Peptide Stapling

One focus of investigations into stapled peptides involves improving protein–protein interaction approaches for the development of anticancer and antiviral drug candidates [31,32,33,34,35,36,37,38,39]. Although stapling has not been widely used for AMP modifications, this technique has shown to be of great value in improving stability, and it is a promising technique for AMP rational design [40]. Despite a rising number of AMPs described, due to a series of drawbacks, only a few natural or non-natural AMPs available on different databases undergo clinical trials. The application of stapling strategies could change this scenario for the better [7].

The Data Repository of Antimicrobial Peptides (DRAMP) contains diverse annotated AMPs and an interesting classification method. Among all 22,259 entries, 5891 are general AMPs (natural and synthetic), 16,110 are from deposited patents, 77 are AMPs in drug development and 181 are stapled AMPs (Figure 2A). Analysis of the 181 stapled peptides revealed that the most common stapling positions are *i*, *i* + 4 or *i, i* + 7, accounting for 82.4% and 16.5%, respectively (Figure 2B). Another observation derived from this analysis is about the most popular stapling strategy, which is all-hydrocarbon stapling, accounting for 89.5% (Figure 2C) [41].

### 2.1. All-Hydrocarbon Stapling

All-hydrocarbon stapling was first established by a research group led by Prof. Gregory Verdine [22]. Inspired by the development of the chemistry for olefinic cross-linking of helices, which is performed through *O*-allyl serine residues via ruthenium-catalyzed ring-closing metathesis (RCM), Verdine’s group designed non-natural amino acids having either *R* or *S* stereochemistry at the α-carbon loaded with tethers of different lengths, followed by RCM and forming the desired clamp at determined peptide positions (Figure 3) [22,23,42]. The olefin tether bridge used for helix stabilization was considered similar to an actual staple, which explains why the term “staple” was introduced in this technique. Stapled helix stabilization relies on the tether length and absolute configuration at the α-carbon position. For this reason, these parameters have been intensely studied. Additionally, it is possible to constrain helices with more than one stapling, independent staples (Figure 1D) or via spiro-bicyclic ring connection, which is also called stitching (Figure 1E) [9,43,44,45].

Regardless of having both single and double stapling strategies available, single stapled peptides stand out in academic research [26,40,46,47]. The success of both strategies has shown to be peptide-dependent, as an increase in the helical content and improved antimicrobial activity is a premature correlation and not always observed to be true (Table 2). Additionally, peptide stapling is often related to increased hemolytic activity, thus highlighting the importance of fine-tuning stapled AMP candidates.

The natural pro-inflammatory chemokine involved in response to microbial infections, CXCL10, is known to be active against different pathogens. However, there is evidence suggesting that CXCL10 have distinct antimicrobial domains. Taking this into account, Crawford et al. [42] mapped the chemokine structure and designed a series of nine overlapping peptides to test against a range of Gram-negative and positive bacteria. Among the derivative peptides, peptide P9, which was originated by the C-terminal α-helix of CXCL10, demonstrated by CD spectroscopy that it requires a specific conducive environment to adopt the native α-helix conformation. That way, in order to promote the secondary structure and enhance the associated antimicrobial activity, P9 was submitted to a stapling at positions 8 and 12, being substituted for the non-natural amino acid (S)-pentenyl alanine. As a result, a great improvement in bactericidal activity was observed, despite the resulting increase in cytotoxicity [48].

All-hydrocarbon stapled analogues of the naturally occurring peptide polybia-MPI (MPI) from the venom of the *Polybia paulista* wasp [49] were synthesized and had their chemical and biological properties evaluated by Luong et al. [47]. The three analogues (MPIS, MPIS-D8N, and MPIS-Q12K) have an oct-4-enyl hydrocarbon staple at the *i*, *i* + 4 positions, tethering amino acid residues 6 and 10, whereas MPIS presented only the stapling.

**Table 2 antibiotics-12-01400-t002:** Stapled peptides reviewed in the present piece and their principal features.

Peptide	Origin	Stapling Style	Amino Acid Substitution	Antimicrobial Activity	Cytotoxicity	Helix Content	Proteolysis Susceptibility	References
P9	CXCL10 chemokine	All-hydrocarbon (*i*, *i* + 4); position 8 and 12	No	Improved	Increased	Increased	NA	[48]
MPIS	Polybia-MPI	All-hydrocarbon (*i*, *i* + 4); position 6 and 10	No	Improved: Gram-positive Indifferent: Gram-negative	Increased	Increased	Decreased	[47]
MPIS-D8N	Polybia-MPI	All-hydrocarbon (*i*, *i* + 4); position 6 and 10	Position 8 D substituted for N	Improved: Gram-positive Indifferent: Gram-negative	Increased	Increased	Decreased	[47]
MPIS-Q12K	Polybia-MPI	All-hydrocarbon (*i*, *i* + 4); position 6 and 10	Position 12 Q substituted for K	Improved: Gram-positive Indifferent: Gram-negative	Increased	Increased	Decreased	[47]
Ac-DS-14W	Non-natural alanine/lysine-based	Double all-hydrocarbon in tandem (*i*, *i* + 4); positions 2 and 6/9 and 13	No	Improved: Gram-positive Indifferent: Gram-negative	Increased	Increased	Decreased	[50]
Ac-DS-5W-	Non-natural alanine/lysine-based (Ac-DS-14W analog)	Double all-hydrocarbon in tandem (*i*, *i* + 4); positions 2 and 6/9 and 13	W insertion at position 5	Improved	Indifferent	Increased	NA	[50]
S-6K-F17	Synthetic peptide 6K-F17	All-hydrocarbon (*i*, *i* + 4); position 10 and 14	No	Improved	Increased	Increased	NA	[40]
S-6K-F17-2G	Synthetic peptide 6K-F17	All-hydrocarbon (*i*, *i* + 4); position 10 and 14	G insertion at positions 8 and 16	Indifferent	Decreased	Indifferent	NA	[40]
S-6K-F17-3G	Synthetic peptide 6K-F17	All-hydrocarbon (*i*, *i* + 4); position 10 and 14	G insertion at positions 8, 13 and 16	Indifferent	Decreased	Indifferent	NA	[40]
S-6K-F17-3GN	Synthetic peptide 6K-F17	All-hydrocarbon (*i*, *i* + 4); position 10 and 14	G insertion at positions 8, 13 and 16 N insertion at position 7	Indifferent	Decreased	Indifferent	NA	[40]
Sau-2	Aurein1.2	All-hydrocarbon (*i*, *i* + 4); position 2 and 6	No	Improved	NA	Increased	Decreased	[46]
Peptide 2	Peptide 1, Mag 2 derivative	All-hydrocarbon (*i*, *i* + 4); position 1 and 5	No	Improved	Decreased	Increased	NA	[26]
Peptide 8	Peptide 1, Mag 2 derivative	All-hydrocarbon (*i, i* + 7); position 8 and 15	No	Decreased	Increased	Increased	NA	[26]
Mag (*i* + 4) 1,15 (A9K)	Mag 2	Double all-hydrocarbon (*i*, *i* + 4); positions 2 and 6/16 and 20	Position 9 A substituted for K	Improved	Decreased	Increased	Decreased	[51]
C-MPI-1	Polybia-MPI	Triazole stapling (*i*, *i* + 4); position 8 and 12	No	Decreased	Increased	Increased	Decreased	[52]
C-MPI-2	Polybia-MPI	Triazole stapling (*i, i + 6*); position 2 and 8	No	Non-active	NA	Indifferent	NA	[52]
Peptide 12 (OH-CM6)	OH-CATH30	Lysine N-alkylation (*i*, *i* + 4); position 12 and 16	No	Improved	Increased, but good therapeutic index	Increased	Decreased	[53]
V26-SP-8	VapC26 α4_54–65_	All-hydrocarbon (*i, i* + 7); position 1 and 8	No	Improved	NA	Increased	NA	[54]
S-TM4 (88–100)	TM4 (88–100)	All-hydrocarbon (*i*, *i* + 4); position 4 and 8	No	Improved	Decreased	Increased	Decreased	[55]

NA: Non-Available.

As a chemical modification, MPIS-D8N and MPIS-Q12K also had single amino acid substitutions. In MPIS-D8N, aspartic acid (D) in position 8 is substituted by asparagine (N), whereas glutamine (Q) 12 is substituted by lysine (K) in MPIS-Q12K. As a result, there was an increase of at least 3-fold in the α-helical contents of all analogues compared to their parent peptide, MPI. However, the antimicrobial activity was not necessarily better enhanced. All three analogues showed improved antibacterial activity between 7- and 23-fold against Gram-positive bacteria; still, there was no improvement for the inhibition of Gram-negative bacteria. The hemolytic assay results revealed that all stapled peptides displayed an increase in hemolytic activity as follows: 12-fold for MPIS-D8N and approximately 4-fold for MPIS and MPIS-Q12K. In the last experiment, MPIS and MPIS-Q12K were submitted to a trypsin digestion assay and showed a 68-fold cleavage decrease [47].

As another example, the non-natural alanine/lysine-based peptide Ac-DS-14W showed two oct-4-enyl staples in tandem, which cross-link residues at positions 2 and 6 for the first staple, and at positions 9 and 13 for the second one. Both staples were incorporated on the same side of the helix so they could be part of the hydrophobic face of an amphipathic helix [50]. Also, to improve the helix stability and amphipathicity, other elements were added, including six lysine residues, four helix-promoting alanine residues, and a tryptophan residue to facilitate interactions with the bacterial membrane. When examined for an α-helical signature by a circular dichroism (CD) spectroscopy, peptide Ac-DS-14W showed enhanced helical content compared to the previous single-stapled (Ac-SS-14W) and unmodified (Ac-UM-14W) analogues. Regarding the antimicrobial activity, an improvement was obtained against Gram-positive bacteria. However, the results were not replicated for Gram-negative bacteria, suggesting that antimicrobial activity is not exclusively correlated with peptide helical conformation [50]. The results of the hemolytic assay revealed that Ac-DS-14W had an increase in hemolysis when compared to the other two analogues. By contrast, when peptides were subjected to trypsin digestion, the double-stapled peptide showed significant stability, not presenting structural alterations even after 60 min of tryptic exposure. Therefore, the peptide sequence was submitted to modifications aiming to increase the antimicrobial activity and reduce the hemolysis of the double-stapled peptide. Three analogues with tryptophan at different positions (positions 3, 5 and 12) were developed, and it was observed that tryptophan at position 5 (which is positioned within the first staple), despite maintaining the levels of hemolysis, considerably increases the antimicrobial activity by 2- to 8-fold against both classes of bacteria.

As described above, hemolysis is an obstacle that must be overcome. For that reason, researchers seeking to develop and design stapled peptides should put effort into finding a well-balanced set of parameters, not only focusing on increasing hydrophobicity and helical content. In this line, Stone et al. [40] used the previously studied synthetic antimicrobial peptide 6K-F17 (KKKKKK-AAFAAWAAFAA-NH_2_) [50,56] to generate four stapled analogues; all of them clamped at the same position, but three of them did so with amino acid substitutions. The stapled S-6K-F17 with no amino acid substitution showed a higher α-helix formation tendency, increased antimicrobial activity against *E. coli* BL21, and a more rapid killing mechanism compared to the non-stapled version (6K-F17) and the other three stapled analogues with amino acid substitutions (S-6K-F17-2G, S-6K-F17-3G and S-6K-F17-3GN). Nevertheless, the observed hemolytic activity of S-6K-F17 was much higher, with an increase varying between 4- and 154-fold. However, the three stapled analogues with higher-polarity amino acid substitutions, namely glycine and/or asparagine, despite not presenting so much helicity, were less toxic to mammalian cells and had only minor antimicrobial activity shifts, 0- to 3-fold. That observed outcome is probably due to a better hydrophobicity balance and selectivity for prokaryotic membranes, since adding staples increases non-specific membrane interactions. These results highlight those constraining helixes, and although they have a high tendency to improve antimicrobial activity, they should not be the only focus in staple studies. Adding polar amino acids, which are unfavorable for helix formation and decrease the helicity index, helps with selectivity and thus with toxicity aspects.

Proteolytic stability is a crucial characteristic for AMP development. Although stapled peptides usually present higher resistance to enzymatic degradation, this property is also dependent on the composition of peptide amino acid residues. Aiming to understand the influence of stability, helicity and antimicrobial activity against pathogenic fungi using an all-hydrocarbon strategy, Zheng et al. [46] developed single-stapled analogues at *i*, *i* + 4 positions using aurein 1.2, a peptide secreted by the Australian bell frog *Litoria aurea* [57]. After CD spectroscopy analyses, most stapled peptides showed an increase in their helical content. However, this increase did not necessarily lead to an improvement in antifungal activity against *Candida* spp. strains. Four stapled analogues showed better antifungal activity and were selected to undergo a protease stability test by chymotrypsin degradation. Data indicate better proteolytic protection within all selected analogues compared to aurein 1.2. However, among these analogues, one presenting a higher helicity index showed lower proteolytic stability. Therefore, the results indicate that, despite achieving a boosted aurein 1.2 analogue, improving a peptide’s antimicrobial potency and stability can be more complex than just constraining helices [46].

Besides all the difficulties that could be found, stapling continues to be a promising strategy for AMP rational design. Hirano et al. [26] decided to improve the potential of a peptide 1 drug, a previously reported Magainin 2 (Mag 2) derivative, by designing and synthetizing 8 side chain stapled peptides, numbered from 2 to 9, in order to evaluate their preferred secondary structures, and their antimicrobial and hemolytic activities. These peptides were synthesized by introducing a suitable (S)-pentenyl alanine and/or (R)-octenyl alanine at the *i*, *i* + 4 or *i, i* + 7 positions, which would lead to side chain stapling. As a result, peptides 2 to 6, presenting *i*, *i* + 4 stapling, showed improved antimicrobial activity, while peptides 7 to 9, presenting *i, i* + 7 stapling, had decreased or indifferent shifts in antimicrobial activity. The investigation of hemolytic activity showed that among all the stapled peptides, only peptide 2 demonstrated hemolytic activity at much higher concentrations compared to MIC, estimated as being 16× higher for gram-positive *S. aureus* and 32× higher for gram-negative *E. coli* and *P. aeruginosa*. All the other peptides presented hemolytic activity at concentrations as low as MIC or even lower, and peptide 8 was the one with the lowest hemolytic concentration. In addition, electrophysiological measurements were performed, revealing that peptide 2 showed higher scores against the DOPE/DOPG membrane than against the DOPC membrane. These results indicate a pore-formation preference for bacterial mimetic than for mammalian mimetic membranes.

In another study, a stapling strategy succeeded in a more promising way [51]. The group developed a predictive algorithm for in silico design of stable, active and membrane-selective stapled AMPs derived from Mag 2 by generating and analyzing libraries from staple-scanning and mutagenesis. First, *i*, *i* + 4 and *i, i* + 7 staple-scanning libraries were generated to test antimicrobial and hemolytic activities, comparing them to Mag 2. After this first step, no clear correlation was observed between peptide potency and hemolysis, since the results were very variable, especially for the *i*, *i* + 4 stapled peptides. After examining biophysical parameters, such as helicity, C18-HPLC retention time, isoelectric point and hydrophobicity, there were still too many variables with no clear correlation. So that more information could be produced, the staple placement along the helix was investigated by a model developed to envision and quantify the hydrophobic interactions along the z axis of peptides, forming hydrophobicity network maps (HNMs). Analyzing the Mag(*i* + 4) library, it became clear that staples placed in hydrophobic patches on the hydrophobic face of Mag 2 had a tendency to increase hemolytic activity, while placing staples in regions that would increase overall hydrophobicity without expanding the hydrophobic panorama on the hydrophobic surface had no effect on increased hemolysis. In addition, after performing lysine scan analysis, there was a decrease in hemolysis when a lysine was substituted at the hydrophobic face, disrupting the hydrophobic network and thus decreasing the number and strength of hydrophobic interactions. However, antimicrobial potency was affected negatively, especially for Gram-positive bacteria, which demanded a more robust hydrophobic interaction to become lysed.

Taking into account all the findings so far, a double-staple scanning library was created and the candidate Mag (*i* + *4*)1,15(A9K) was selected for synthesis and characterization. The insertion of the two clamps in the *i*, *i* + 4 position provided a predominantly α-helical structure and protection against the action of proteinase K. Regarding the antimicrobial activity, the double-stapled peptide had 2-fold greater activity against *E. coli* and 4-fold greater activity against *P. aeruginosa* when compared to the single-stapled one. In addition, the peptide presented no hemolytic activity in the effective dose range. On the other hand, in vivo tests showed renal toxicity. However, after reducing hydrophobicity by mutating norleucine to alanine at position 21 and adding compensatory C-terminal positive charges (N22K and S23K), the renal toxicity was resolved.

Understanding the possible drawbacks related to characterizing stapled peptides, the importance of studies that pursue more complex and rich evaluations is clear, especially those taking advantage of computational approaches. Thus, it can be possible to generate better knowledge and insights on how to improve stapled peptides’ therapeutics and competitiveness.

### 2.2. Alternative Stapling Strategies

Although it is the most used stapling technique, hydrocarbon stapling is only one of the many approaches available. The term “stapling” has become a general term to refer to cyclic peptides, especially those that aim to constrain helices [19]. Alternatively, there are one-component approaches such as disulfide or amide bonding, and two-component methodologies, including thioether formation and alkylation of Lys residues [17]. In this perspective, Liu et al. [52], also using polybia-MPI as the original counterpart, generated the stapled analogues C-MPI-1 and C-MPI-2 by a triazole stapling technique. From these, only C-MPI-1 presented a slight α-helical structure. Even though C-MPI-1 showed higher helicity content, there was little or no enhancement in antimicrobial activity against Gram-positive and Gram-negative bacteria and worse hemolysis results compared to MPI.

Similarly, the cationic AMP OH-CM6, derived from the natural cathelicidin peptide OH-CATH30 from the king cobra *Ophiophagus hannah* was used to generate stapled analogues by N-alkylation of lysine [53]. In this strategy, the lysine residues on the hydrophilic face of the native peptide were clamped by an alkyl linker at *i*, *i* + 4 and *i, i* + 7. During this process, ten stapled peptides were generated and screened for antimicrobial activity against Gram-positive and Gram-negative bacteria, including clinical isolated strains. The best candidate (peptide 10) was chosen after antimicrobial screening. From this selected peptide, different analogues were synthesized with an aralkyl linker. One of these peptides (peptide 12) presented antibacterial activity against methicillin-resistant *S. aureus* in concentrations comparable to Vancomycin. In addition to that, despite the toxicity observed in the assay against the HEK 293T cell line, the concentration was 32-fold higher than the observed minimum inhibitory concentration, showing an interesting therapeutic index. Moreover, the α-helical content was around 45% and the proteolytic stability was better than other analogues tested, namely the linear and one of the stapled analogues, indicating a promising molecule for further investigation.

## 3. Stapling for Protein–Protein Interaction Targets

There is a large amount of data on stapled peptides for protein–protein interaction mechanisms [58]. These studies mostly focus on different disorders including cancer, viral infections and cardiovascular diseases. The explanation for this is that PPIs mediate cellular functions, molecular and signal pathways. Finding critical interaction domains allows for the rational design of PPI inhibitors [24,59,60]. Matching the antimicrobial resistance context and protein–protein interaction strategies, Kang et al. [54] took advantage of the toxin–antitoxin (TA) system VapBC26 from *Mycobacterium tuberculosis* to develop an AMP agent that can act as an inhibitor of intermolecular protein–protein interactions. Firstly, non-crucial residues were selected for stapling modification, and the stapled peptide V26-SP-8 was synthesized and derived from VapC26 α4_54–65_, previously described as inhibiting the VapBC26 TA system. The stapled peptide mimics the VapC26 antitoxin-binding region, competing for the binding site in VapB26. Once VapC26 is free, the active site is exposed and induces the digestion of bacterial rRNA. The stapled peptide V26-SP-8 presented a higher helical content, greater activity, improved VapB26 binding affinity and cell internalization compared to the linear counterpart VapC26 α4_54–65_. Therefore, V26-SP-8 could be a promising therapeutic molecule against multidrug-resistant tuberculosis, and it indicates a new point of view for the establishing of targets for antibiotic development.

An application of peptide stapling to a membrane-embedded target to block and disrupt particular protein–protein interactions and, consequently, protein function [55] has been reported. As well as combating efflux pump-expressing multidrug-resistant bacteria, this functionality may be very useful for cancer cells that are multidrug resistant. It could transport chemotherapeutics using efflux pumps from the ATP binding cassette, or to G-protein-coupled oncogenic receptors that automatically signal after dimerization. In this study, a stapled peptide that inhibits specific efflux and resensitizes Hsmr-expressing *E. coli* cells to ethidium bromide, while being non-toxic to mammalian blood cells, was designed from the oligomerization of the TM4-TM4 site in the bacterial efflux pump Hsmr. A set of peptides stapled at positions 92 and 96 (S-TM4 (88–100)) were produced as a result of the helix stapling of TM4 (88–100), which was chosen because it was the minimal inhibitory motif. This was accomplished by adding alkenylalanyl residues at the *i* and *i* + 4 positions, followed by Grubbs metathesis [42].

In another example for disrupting protein–protein interactions, in order to increase the α-helicity, stability and biological activity of peptides in vitro and in vivo, Walensky et al. [61] used synthetic, α,α-disubstituted amino acids with olefin-containing tethers to create an all-hydrocarbon staple using ruthenium-catalyzed olefin metathesis. The stabilized alpha helix of BCL-2 domains (SAHBs), a group of hydrocarbon-stapled peptides, was created to resemble the BH3 domain of BID, which is a pro-apoptotic protein exclusive to BH3 that links intrinsic and extrinsic apoptotic pathways in response to death receptor stimulation. Anti-apoptotic proteins (like BCL-2 and BCL-XL) can bind to and sequester activated BID (tBID), but they can also activate the multidomain pro-apoptotic proteins BAX and BAK, releasing cytochrome c and activating the mitochondrial apoptosis mechanism.

Similarly, there is interest in the ability of peptides to inhibit protein–protein interactions [62], and hydrophobic cross-links were introduced to increase the peptide binding affinities significantly for ExoS–14-3-3, a component of the virulence system of the pathogenic bacterium *Pseudomonas aeruginosa*. Three peptides were first created, each having two non-natural amino acids connected by a (CH_2_)_8_ bridge. Two *S*-configured non-natural amino acids (X_S_) were present in one peptide (a_SS_8) at positions *i* and *i* + 4. The *R*- and *S*-configured building blocks (X_R_ and X_S_, respectively) were added to the other two peptides (β- and γ_RS_8) at positions *i* and *i* +3, respectively.

Using protein–protein interactions as targets for the development of new drugs has broadened the alternatives used to fight antimicrobial resistance with more effective and specific strategies [63,64]. The range of the structural requirements of protein–protein interfaces make it extremely challenging to find suitable selective inhibitors through the screening of conventional small-molecule libraries [64]. In fact, peptides are more potent and selective in terms of binding to protein–protein interaction sites compared to small molecules. Besides that, peptides are less toxic, since they are degraded into amino acids, while small molecules often produce toxic metabolites [30]. An alternative strategy is the rational creation of inhibitors using peptide-binding epitopes generated from protein–protein complexes. Most instances of biologically active mimetics use stapled peptides. Various cross-link designs have been created, particularly for α-helices. According to recent comparison investigations, including the ones above, several different cyclization techniques can deliver helical stapled peptides with equivalent target affinity enhancements, but it is still necessary to thoroughly assess how these cross-linking techniques affect the bioavailability of the resultant peptides [64].

## 4. Conclusions

The introduction of chemical modifications and non-natural amino acids has shed new light on peptide-based drug research by overcoming limitations on peptide therapeutics. Stabilizing helices is one of the outcomes most explored in chemical modification methodologies, since there is a strong correlation between helices and peptide activity. Stapled antimicrobial peptides appear to be an auspicious class of antimicrobial agents, enabling an increase in antimicrobial activity and structural stability of natural and non-natural AMPs. Despite this, the stapling technique has some limitations itself. The increase in hydrophobicity derived from the stapling [18] usually results in reduced solubility in aqueous solutions and often leads to an increase in hemolysis [51]. Another topic for discussion regarding stapled peptides should be about their selectivity. Throughout the studies presented in this review, stapled peptides often acquired greater toxicity. Since stapled peptides compared to linear ones do not usually present lower activity against microorganisms but less selectivity, we could hypothesize that structural plasticity is an important feature for AMPs.

It is important to highlight the role of computational approaches for staple design. Most studies are focused on examining stapling positioning by biophysical characterization of every possible construct on the same peptide. Yet, taking this road does not provide rich and comprehensive insights into stapled peptides’ constructs, in addition to being expensive and time-consuming [30]. Therefore, there is no guideline to help in choosing a convenient base architecture. Contributing to this, many studies do not refine the investigation towards alternatives to selected basic architectures [43]. This refinement would help investigators better understand whether there are collateral effects arising from the usage of stapled AMPs other than toxicity and balance parameters like helicity and hydrophobicity to develop more promising molecules.

In contrast, there are multiple different possibilities for using stapling in the rational design of AMPs. As shown here, direct killing mechanisms by membrane disruption are the focus of cationic stapled AMPs and are often reported as a mode of action that is less prone to acquired resistance [51]. However, taking advantage of protein–protein interaction strategies to kill bacteria could be a promising solution. Despite being a technique applied mostly to diseases such as cancer and viral infections, it can be used to expand possible bacterial targets, other than the cytoplasmatic membrane [54]. In a similar matter, designing blockers of resistance mechanisms such as the efflux pump could be a viable approach [55].

Taken together, this review focuses on how stapled peptides have been explored for use as antimicrobials and encourages further studies related to this, thereby improving the suitability of AMPs for the competitive market.

## Figures and Tables

**Figure 1 antibiotics-12-01400-f001:**
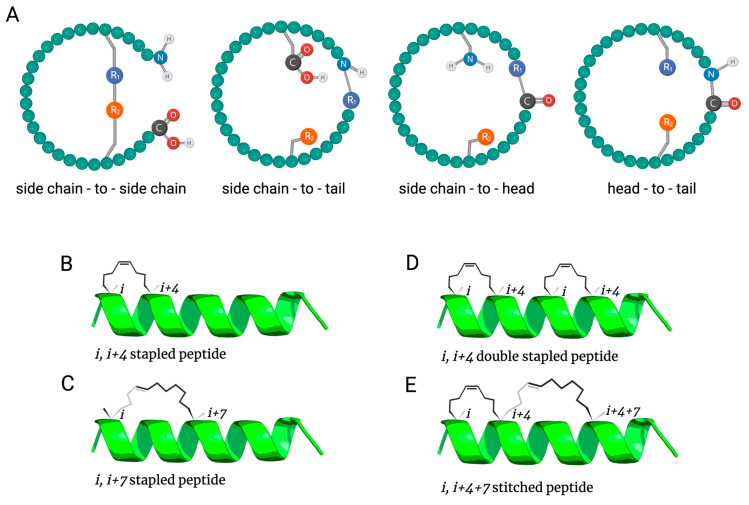
Cyclization and stapling strategies. Cyclization strategies (**A**) varying the linked positions in the peptide structure and stapling (**B***–***E**), which is a type of side chain-to-side chain cyclization, in which the most common positions are *i*, *i* + 4 (**B**) and *i*, *i* + 7 (**C**) for a single staple and possible positions for double stapling (**D**) and stitching (**E**).

**Figure 2 antibiotics-12-01400-f002:**
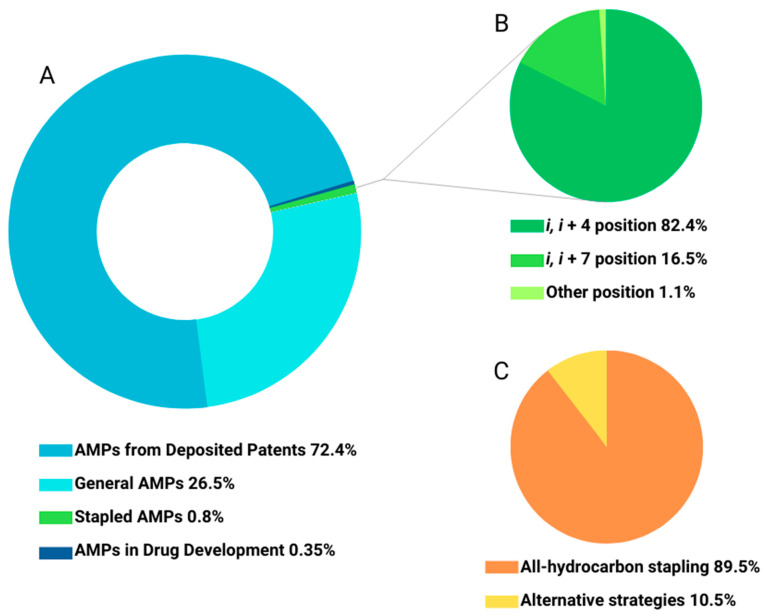
Classification of AMPs deposited on the Data Repository of Antimicrobial Peptides (DRAMP) describing (**A**) type of AMP, (**B**) most common positions for stapled AMPs and (**C**) stapling strategy applied.

**Figure 3 antibiotics-12-01400-f003:**
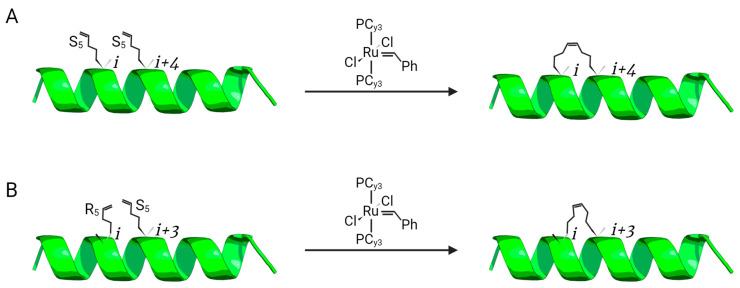
Cross-linking mediated by ruthenium-catalyzed ring-closing metathesis (RCM) between non-natural amino acids presenting S (**A**) and/or R (**B**) stereochemistry forming the staple at the desirable position within the α-helix structure.

**Table 1 antibiotics-12-01400-t001:** AMPs’ most common modifications and their principal advantages. ↑: increased; ↓: decreased.

Modification Type	Advantages
**INSERTION OF D-AMINO ACIDS**	↑ proteolytic stability ↑ membrane insertation
**PEGYLATION**	↑ proteolytic stability ↓ toxicity ↑ biocompatibility ↑ plasma half-live
**ACETYLATION**	↑ proteolytic stability
**DIMERIZATION**	↑ antimicrobial activity ↑ membrane interaction ↑ membrane permeability
**LIPIDATION**	↑ antimicrobial activity ↑ proteolytic stability ↑ membrane permeability ↑ bioavailability
**CYCLIZATION**	↑ antimicrobial activity ↑ proteolytic stability

## Data Availability

No new data were created or analyzed in this study. Data sharing is not applicable to this article.

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
