# Peer review of "Peptide Stapling Applied to Antimicrobial Peptides"

_antibiotics, 2023, doi:10.3390/antibiotics12091400_

Round 1

Reviewer 1 Report

The review "Peptide stapling applied to antimicrobial peptides", devoted to structural modification of antimicrobial peptides and related changes in their biological activity, makes a favorable impression. It may be of interest to a wide range of specialists in the study of antimicrobial peptides.

At the same time, there are a number of remarks to the content. It would be desirable to give a more complete characterization of the described peptides, in particular their source. For example:

polybia-MPI peptide from the venom of the social wasp Polybia paulista

Ac-DS-14W non-natural alanine/lysine-based peptide

6K-F17 – artificial alanine/lysine-rich peptide

Aurein1.2 antimicrobial peptide secreted by the Australian tree frog Litoria aurea

OH-CATH30 – cathelicidin peptide from the king cobra

It could be noted that no studies on the modification of mammalian peptides have been described; therefore, an article could be recommended for consideration:

Smirnova M.P., Kolodkin N.I., Kolobov A.A., Afonin V.G., Afonina I.V., Stefanenko L.I., Shpen’ V.M., Shamov O.V., Kolobov A.A. Indolicidin analogs with broad-spectrum antimicrobial activity and low hemolytic activity. Peptides 2020, 132, 170356, doi:10.1016/j.peptides.2020.170356

The authors have not avoided some mistakes:

1) Line 268 – ABC binding cassette

Comment – Actually, ATP-binding cassette (ABC)

2) Line 3122.2. Alternative stapling strategies

Comment – Actually, 2.3. Alternative stapling strategies

3) Line 324 – Similarly, the cationic AMP OH-CM6, derived from the natural peptide OH-CATH30, was used to generate stapled analogues by N-alkylation of lysine [53].

Comment – There is no mention of the OH-CATH30 in the review under reference [53]

4) Line 336 – proteolytic activity was better

Comment – Apparently, proteolytic stability was better

The References section needs serious correction as there is a lot of duplication:

References 11 and 53 are the same article, as well 15 and 35, 23 and 52, 36 and 47, 41 and 43, 44 and 55, 46 and 56.

Author Response

Dear Reviewer 1,

First, we would like to thank you for the time and effort put into this piece of work. We appreciate all the suggestions and contributions to the manuscript. Below, you can find answers and comments to your suggestions, point by point. All changes are highlighted in the new version of the manuscript.

Reviewer 1

  • “ It would be desirable to give a more complete characterization of the described peptides, in particular their source”

 Response: The origin of the peptides reviewed was added to their description.

  • “Icould be noted that no studies on the modification of mammalian peptides have been described; therefore, an article could be recommended for consideration:

Smirnova M.P., Kolodkin N.I., Kolobov A.A., Afonin V.G., Afonina I.V., Stefanenko L.I., Shpen’ V.M., Shamov O.V., Kolobov A.A. Indolicidin analogs with broad-spectrum antimicrobial activity and low hemolytic activity. Peptides 2020, 132, 170356, doi:10.1016/j.peptides.2020.170356”

Response: A study using peptides from mammalian origin was added. Despite the reviewer’s suggestion of Smirnova et al. 2020, which we appreciate, we opted to include a more recent manuscript to represent mammalian peptide stapling.

Crawford, M. A., Ward, A. E., Gray, V., Bailer, P., Fisher, D. J., Kubicka, E., Cui, Z., Luo, Q., Gray, M. C., Criss, A. K., Lum, L. G., Tamm, L. K., Letteri, R. A., & Hughes, M. A. (2023). Disparate Regions of the Human Chemokine CXCL10 Exhibit Broad-Spectrum Antimicrobial Activity against Biodefense and Antibiotic-Resistant Bacterial Pathogens. ACS infectious diseases, 9(1), 122–139. https://doi.org/10.1021/acsinfecdis.2c00456.

  • “The authors have not avoided some mistakes:”

    1) Line 268 – ABC binding cassette

    Comment – Actually, ATP-binding cassette (ABC)

    2) Line 312 – 2.2. Alternative stapling strategies

    Comment – Actually, 2.3. Alternative stapling strategies

    3) Line 324 – Similarly, the cationic AMP OH-CM6, derived from the natural peptide OH-CATH30, was used to generate stapled analogues by N-alkylation of lysine [53].

    Comment – There is no mention of the OH-CATH30 in the review under reference [53]

    4) Line 336 – proteolytic activity was better

    Comment – Apparently, proteolytic stability was better

Response: All mistakes listed were corrected in the new version. We appreciate the suggestions.

  • “The References section needs serious correction as there is a lot of duplication: References 11 and 53 are the same article, as well 15 and 35, 23 and 52, 36 and 47, 41 and 43, 44 and 55, 46 and 56.”

 Response: An error with the reference software led to the duplication and was not corrected before submission. We apologize for that. We reviewed and corrected all references before submission for this new version.

Reviewer 2 Report

This manuscript is a review about the application of the stapling technique as a strategy to stabilize the active structure of antimicrobial peptides. The authors explain different variants of peptide stapling, both by connectivity and chemical mechanism, and they include and discuss several examples of recent works of the main contributors to this field.

The manuscript gives a general idea of the state-of-the-art of the stapling strategies applied to the improvement of the therapeutic index of AMPs. However, I strongly recommend some changes in the structure, expand the contents and include additional figures/tables, since I think the contribution of this manuscript to the field could be overshadowed by other already published reviews (see Moiola et al., 2019).

I would like to share with the authors some comments on the manuscript:

1- I suggest to expand the bibliographic research, add some additional references or consider more recent ones. For instance:

- Verdine, G. L., & Hilinski, G. J. (2012). Stapled peptides for intracellular drug targets. In Methods in enzymology (Vol. 503, pp. 3-33). Academic Press.

- Cromm, P. M., Spiegel, J., & Grossmann, T. N. (2015). Hydrocarbon stapled peptides as modulators of biological function. ACS chemical biology, 10(6), 1362-1375.

- Moiola, M., Memeo, M. G., & Quadrelli, P. (2019). Stapled peptides—a useful improvement for peptide-based drugs. Molecules, 24(20), 3654.

- Li, X., Chen, S., Zhang, W. D., & Hu, H. G. (2020). Stapled helical peptides bearing different anchoring residues. Chemical Reviews, 120(18), 10079-10144.

- Robertson, N. S., & Jamieson, A. G. (2015). Regulation of protein–protein interactions using stapled peptides. Reports in Organic Chemistry, 65-74.

- Xie, X., Gao, L., Shull, A. Y., & Teng, Y. (2016). Stapled peptides: providing the best of both worlds in drug development. Future Medicinal Chemistry, 8(16), 1969-1980.

2- Section 2 could be enriched by adding more figures or tables. Reviews are interesting because they summarize published data; figures and tables facilitate its visualization. For example, the paragraph encompassing lines 91 to 97 could be represented in a figure, like a piechart. Likewise, a table containing the main information of the different examples mentioned about published studies of stapled AMPs will help to have a quick vision of the state-of-the-art. Please, consider this type of visual enrichment, some of the references I mentioned before can help as inspiration. 

3- Please, reconsider section organization as section 2 (Antimicrobial peptide stapling) is subdivided in three subsections (2.1, 2.2 and 2.3), which in principle correspond to different stapling strategies: all-hydrocarbon, stapling for PPI targets, and alternative stapling strategies. However, I feel that subsection 2.2 is misplaced here, because subsections 2.1 and 2.3 indeed talk about stapling strategies, but subsection 2.2 talks more about a strategy for the application of the stapled peptides. I think that maybe you can dedicate a subsection to stapling strategies (including all-hydrocarbon and alternative strategies) and another subsection to application strategies of stapled peptides (including their direct use as AMPs or their use as inhibitors/disruptors of the protein-protein interactions). This organization will bring more clarity and coherence to the manuscript, improving its readability.

4 - Just a minor comment: in line 194 you mention "Magainin 2 (Mag 2)", and in line 212 you mention it again but you write "magainin II (Mag 2)". Please, be coherent with the spelling and I think that if you abbreviate the name as Mag 2 the first time, the second time you can directly use that abbreviation "Mag 2".

The quality of English language in this manuscript is good.

Author Response

Dear Reviewer 2,

First, we would like to thank you for the time and effort put into this piece of work. We appreciate all the suggestions and contributions to the manuscript. Below, you can find answers and comments to your suggestions, point by point. All changes are highlighted in the new version of the manuscript.

  • “I suggest to expand the bibliographic research, add some additional references or consider more recent ones. For instance:”

Response: We considered and accepted the bibliographic suggestion. You can find them within the text and reference list.

  • “Section 2 could be enriched by adding more figures or tables. Reviews are interesting because they summarize published data; figures and tables facilitate its visualization. For example, the paragraph encompassing lines 91 to 97 could be represented in a figure, like a piechart. Likewise, a table containing the main information of the different examples mentioned about published studies of stapled AMPs will help to have a quick vision of the state-of-the-art. Please, consider this type of visual enrichment, some of the references I mentioned before can help as inspiration.”

Response: We added 2 extra figures and a table to the manuscript.

  • “Please, reconsider section organization as section 2 (Antimicrobial peptide stapling) is subdivided in three subsections (2.1, 2.2 and 2.3), which in principle correspond to different stapling strategies: all-hydrocarbon, stapling for PPI targets, and alternative stapling strategies. However, I feel that subsection 2.2 is misplaced here, because subsections 2.1 and 2.3 indeed talk about stapling strategies, but subsection 2.2 talks more about a strategy for the application of the stapled peptides. I think that maybe you can dedicate a subsection to stapling strategies (including all-hydrocarbon and alternative strategies) and another subsection to application strategies of stapled peptides (including their direct use as AMPs or their use as inhibitors/disruptors of the protein-protein interactions). This organization will bring more clarity and coherence to the manuscript, improving its readability.”

Response: We appreciate the suggestion. Sections were reorganized. Subsection 2.2 Stapling for PPI targets was transformed into a new section 3. This way, section 2 corresponds to structural stapling diversity (2.1 All-hydrocarbon and 2.2 Alternative stapling), and section 3 is dedicated to this specific approach of using stapled peptides as PPI inhibitors.

  • “Just a minor comment: in line 194 you mention "Magainin 2 (Mag 2)", and in line 212 you mention it again but you write "magainin II (Mag 2)". Please, be coherent with the spelling and I think that if you abbreviate the name as Mag 2 the first time, the second time you can directly use that abbreviation "Mag 2.”

 Response: We agree with the reviewer and we corrected the spelling.

Reviewer 3 Report

The review by Lourenço et al. provides an insightful overview of the application of peptide stapling in the context of antimicrobial drug discovery. The authors first discuss different strategies for peptide stapling and cyclization, followed by their interpretation of several research studies on the topic. The authors did a good job in presenting key observations of these studies in a simple manner. I appreciate the authors efforts in summarizing recent advances in the field. Overall, I am enthusiastic about the manuscript, and believe that the content will be useful to readers who wish to learn about peptide stapling for antimicrobial peptides. I indicated ‘moderate’ for contribution to the field as a review inherently does not deal with new discoveries. I recommend publishing the review with the following changes:

PXLY = Page X, Line Y

1.     The classification discussed under section 2 is confusing. Section 2.1 is named “All-hydrocarbon stapling”, while section 2.2 (Stapling for protein-protein interaction targets) also discusses few peptides with all-hydrocarbon stapling (P6L280). Instead, the titles may be re-written with a focus on the mode of action – for example, the title of Section 2.1 can be rephrased to “Stapling for membranolytic antimicrobial peptides” or something in similar lines. This will be consistent with the introduction, as the authors have already talked about membranolytic and non-membranolytic modes of action (P1L34).

2.     P2L58: proteolytic molecule can be rephrased as ‘a molecule less susceptible to proteolysis’. Reason: if proteolytic enzymes are enzymes which cleave peptide bonds, then proteolytic molecules are those molecules which are capable of cleaving peptide bonds.

3.     P2L78: References 20-23 have been cited inappropriately. This specific sentence does not require any citations.

Optional changes:

4.     P3L82: It may be easier for the readers of the manuscript to understand the different types of peptide stapling using schematics. For instance, a reaction scheme with reactant and product for ring closing metathesis on a peptide for all-hydrocarbon stapling. Similarly, alternative stapling strategies may also be explained in a figure using simple chemical schematics.

5.   A new study may be included in the discussion/references, which was published a couple of months ago. https://doi.org/10.1021/acs.jmedchem.3c00140  

Author Response

Dear Reviewer 3,

First, we would like to thank you for the time and effort put into this piece of work. We appreciate all the suggestions and contributions to the manuscript. Below, you can find answers and comments to your suggestions, point by point. All changes are highlighted in the new version of the manuscript.

  • “The classification discussed under section 2 is confusing. Section 2.1 is named “All-hydrocarbon stapling”, while section 2.2 (Stapling for protein-protein interaction targets) also discusses few peptides with all-hydrocarbon stapling (P6L280). Instead, the titles may be re-written with a focus on the mode of action – for example, the title of Section 2.1 can be rephrased to “Stapling for membranolytic antimicrobial peptides” or something in similar lines. This will be consistent with the introduction, as the authors have already talked about membranolytic and non-membranolytic modes of action (P1L34).”

Response: We’d like to thank the reviewer for the input. Sections were reorganized to improve readability. Subsection 2.2 Stapling for PPI targets was transformed into a new section 3. This way, section 2 corresponds to structural stapling diversity (2.1 All-hydrocarbon and 2.2 Alternative stapling), and section 3 is dedicated to this specific approach of using stapled peptides as PPI inhibitors.

  • “P2L58: proteolytic molecule can be rephrased as ‘a molecule less susceptible to proteolysis’. Reason: if proteolytic enzymes are enzymes which cleave peptide bonds, then proteolytic molecules are those molecules which are capable of cleaving peptide bonds.”

 Response: We agree with the reviewer and have changed the sentence in the manuscript.

  • “P2L78: References 20-23 have been cited inappropriately. This specific sentence does not require any citations.”

 Response: The idea was to offer the reader pieces of literature that would cover what we don´t in this review, such as the chemical aspects of different stapling strategies. We understand the sentence wasn´t very clear, so we have corrected the sentence and kept the citations.

  • “P3L82: It may be easier for the readers of the manuscript to understand the different types of peptide stapling using schematics. For instance, a reaction scheme with reactant and product for ring-closing metathesis on a peptide for all-hydrocarbon stapling. Similarly, alternative stapling strategies may also be explained in a figure using simple chemical schematics.”

Response: A schematic of the all-hydrocarbon stapling reaction was added in Figure 2.

  • “A new study may be included in the discussion/references, which was published a couple of months ago. https://doi.org/10.1021/acs.jmedchem.3c00140” 

Response: We focused on alpha-helix stabilization other than beta-sheet, however, we appreciate the suggestion and added it in the text, highlighting that stapling is not exclusively to constrain helices.

Round 2

Reviewer 2 Report

I would like to thank the authors for the changes introduced in the manuscript upon reviewer's suggestions. The text is now more consolidated, with a better organization and readability. 

Yet, I found some minor issues that I would like to share with the authors, including suggestions and comments:

Line 18 - Suggestion: "This technique" change by "The latter one" or a similar expression to make clear that you refer to the last one of the mentioned approaches, the all-hydrocarbon stapling.

Line 40 - Suggestion: I would change "undergo proteolytic activity" by "undergo proteolytic degradation".

Line 43 - Suggestion: "Therefore, chemical modification of AMPs are essentially oriented to improve their instability and antimicrobial activity and decrease cytotoxicity."

Line 47 - Suggestion: It would be interesting to briefly pinpoint how every approach improves the AMPs properties and why peptide cyclization may be more promising than the others.

Line 50 - Comment: This is useful when the restrained conformation is the most efficient to exert the peptide function, because it has been well-established in many works that precisely peptide flexibility is often more crucial for the interaction with membranes that the presence of stable or rigid secondary structures. You point to this in your comment in line 127 and in lines 172/173: the correlation between helical content and antimicrobial acticivity is not always observed, because sometimes a more flexible, random coil structure is able to establish a more efficient interaction thanks to its conformational adaptability.

Line 54 - "Figure 1a" should be substituted by "Figure 1A"

Line 58 - "Figure 1" should be substituted by "Figure 1B"

FIGURE 1A - Maybe I am mistaken, but I think that the side chain to tail and side chain to head panels are not well represented. When you say head and tail, I think that you refer to the N- and the C-terminus of the peptide, respectively. So, in the side chain to tail panel, the side chain should be linked to the tail (the C-terminus), which must be at the end of the chain. Analogously, in the side chain to head, the side chain should be linked to the head (the N-terminus), which must be at the begining ofthe chain. 

FIGURE 1 caption - Where you say "and, possibly, positions for double stapling (D) and stitching (E)", I think it sholud be "and possible position for double stapling (D) and stitching (E)"

FIGURE 2 - If you use only one decimal position for the percentages, use it for every amount (you use only one decimal position, but in AMPs in drug development, you use two decimal positions). Use one or two decimal positions, but always the same. 

Line 135 - It should say "derivative" instead "derivatives"

Line 141 - Change "despite consequence increase in cytotoxicity" by "despite the resulting increase in cytotoxicity".

Line 142 - Change "All-hydrocarbon stapled analogues of the naturally occurring peptide, from the venom of Polybia paulista wasp, polybia-MPI (MPI)" by "All-hydrocarbon stapled analogues of the naturally occurring peptide polybia-MPI (MPI), from the venom of Polybia paulista wasp,"

TABLE 1 - Please write gram with capital G "Gram" in the antimicrobial activity column.

Line 164 - I don't understand "to obtain the hydrophobic face of an amphipathic helix", do you mean that they staple those residues to make that region of the helix more hydrophobic and make the whole helix amphipathic?

Line 256 - Change "gram-positive" by "Gram-positive"

Line 264 - I will remove "against red blood cells" because it is implicit in the term hemolysis.

Line 271 - Change "especially ones" by "especially that ones"

Line 288 - You can include the scientific name of the king cobra, since you included it when you mentioned other species before.

Line 328 - it should be "the oligomerization of the TM4-TM4 site"

Line 334 - In vitro and in vivo should be in italics.

Author Response

Dear Reviewer 2,

We appreciate the quick response and the new considerations to improve our manuscript. Below are our answers and comments for the suggestions made in the second round. All changes from the first round are highlighted in yellow as all changes from the second round are highlighted in light blue.

Line 18 - Suggestion: "This technique" change by "The latter one" or a similar expression to make clear that you refer to the last one of the mentioned approaches, the all-hydrocarbon stapling.

Response: Accepted and corrected.

Line 40 - Suggestion: I would change "undergo proteolytic activity" by "undergo proteolytic degradation".

Response: Accepted and corrected.

Line 43 - Suggestion: "Therefore, chemical modification of AMPs are essentially oriented to improve their instability and antimicrobial activity and decrease cytotoxicity."

Response: Accepted.

Line 47 - Suggestion: It would be interesting to briefly pinpoint how every approach improves the AMPs properties and why peptide cyclization may be more promising than the others.

Response: A table showing peptides modifications and their advantages was added to cover the suggested tòpic (Table 1).

Line 50 - Comment: This is useful when the restrained conformation is the most efficient to exert the peptide function, because it has been well-established in many works that precisely peptide flexibility is often more crucial for the interaction with membranes that the presence of stable or rigid secondary structures. You point to this in your comment in line 127 and in lines 172/173: the correlation between helical content and antimicrobial acticivity is not always observed, because sometimes a more flexible, random coil structure is able to establish a more efficient interaction thanks to its conformational adaptability.

Response: A new sentence was added to make it clearer: Nevertheless, further in this review, flexibility aspects are discussed as being an interesting allied for more efficient interactions.

Line 54 - "Figure 1a" should be substituted by "Figure 1A"

Response: Accepted and corrected.

Line 58 - "Figure 1" should be substituted by "Figure 1B"

Response: Since the citation is about stapling in general, the citation was changed for Figure 1B-E. 

FIGURE 1A - Maybe I am mistaken, but I think that the side chain to tail and side chain to head panels are not well represented. When you say head and tail, I think that you refer to the N- and the C-terminus of the peptide, respectively. So, in the side chain to tail panel, the side chain should be linked to the tail (the C-terminus), which must be at the end of the chain. Analogously, in the side chain to head, the side chain should be linked to the head (the N-terminus), which must be at the begining ofthe chain. 

Response: Accepted and corrected.

FIGURE 1 caption - Where you say "and, possibly, positions for double stapling (D) and stitching (E)", I think it sholud be "and possible position for double stapling (D) and stitching (E)"

Response: Accepted and corrected.

FIGURE 2 - If you use only one decimal position for the percentages, use it for every amount (you use only one decimal position, but in AMPs in drug development, you use two decimal positions). Use one or two decimal positions, but always the same. 

Response: Accepted and corrected.

Line 135 - It should say "derivative" instead "derivatives"

Response: Accepted and corrected.

Line 141 - Change "despite consequence increase in cytotoxicity" by "despite the resulting increase in cytotoxicity".

Response: Accepted and corrected.

Line 142 - Change "All-hydrocarbon stapled analogues of the naturally occurring peptide, from the venom of Polybia paulista wasp, polybia-MPI (MPI)" by "All-hydrocarbon stapled analogues of the naturally occurring peptide polybia-MPI (MPI), from the venom of Polybia paulista wasp,"

Response: Accepted and corrected.

TABLE 1 - Please write gram with capital G "Gram" in the antimicrobial activity column.

Response: Accepted and corrected. Now, this would be Table 2.

Line 164 - I don't understand "to obtain the hydrophobic face of an amphipathic helix", do you mean that they staple those residues to make that region of the helix more hydrophobic and make the whole helix amphipathic?

Response: Yes. The statement was rewritten to improve readbility: Both staples were incorporated on the same side of the helix so they could be part of the hydrophobic face of an amphipathic helix.

Line 256 - Change "gram-positive" by "Gram-positive"

Response: Accepted and corrected.

Line 264 - I will remove "against red blood cells" because it is implicit in the term hemolysis.

Response: Accepted and corrected.

Line 271 - Change "especially ones" by "especially that ones"

Response: Accepted and corrected for especially those ones.

Line 288 - You can include the scientific name of the king cobra, since you included it when you mentioned other species before.

Response: Accepted.

Line 328 - It should be "the oligomerization of the TM4-TM4 site"

Response: Accepted and corrected.

Line 334 - In vitro and in vivo should be in italics.

Response: Accepted and corrected.